# An Early Disturbance in Serotonergic Neurotransmission Contributes to the Onset of Parkinsonian Phenotypes in *Drosophila melanogaster*

**DOI:** 10.3390/cells11091544

**Published:** 2022-05-05

**Authors:** Rafaella V. Zárate, Sergio Hidalgo, Nicole Navarro, Daniela Molina-Mateo, Duxan Arancibia, Francisca Rojo-Cortés, Carlos Oliva, María Estela Andrés, Pedro Zamorano, Jorge M. Campusano

**Affiliations:** 1Departamento de Biología Celular y Molecular, Facultad de Ciencias Biológicas, Pontificia Universidad Católica de Chile, Santiago 8330025, Chile; nenavarro@uc.cl (N.N.); dfmolina@uc.cl (D.M.-M.); frrojo@uc.cl (F.R.-C.); colivao@bio.puc.cl (C.O.); mandres@bio.puc.cl (M.E.A.); 2Departamento Biomédico, Facultad de Ciencias de la Salud, Universidad de Antofagasta, Antofagasta 1240000, Chile; duxan.arancibia@uantof.cl (D.A.); zamorano@gmail.com (P.Z.); 3Department of Entomology and Nematology, College of Agricultural and Environmental Sciences, University of California, Davis, CA 95616, USA; shidalgo@ucdavis.edu; 4Instituto Antofagasta, Universidad de Antofagasta, Antofagasta 1240000, Chile; 5Centro Interdisciplinario de Neurociencia UC, Santiago 8331150, Chile

**Keywords:** Parkinson’s disease, *Drosophila melanogaster*, PINK1, pre-symptomatic phase, serotonin, SERT

## Abstract

Parkinson’s disease (PD) is a neurodegenerative disease characterized by motor symptoms and dopaminergic cell loss. A pre-symptomatic phase characterized by non-motor symptoms precedes the onset of motor alterations. Two recent PET studies in human carriers of mutations associated with familial PD demonstrate an early serotonergic commitment—alteration in SERT binding—before any dopaminergic or motor dysfunction, that is, at putative PD pre-symptomatic stages. These findings support the hypothesis that early alterations in the serotonergic system could contribute to the progression of PD, an idea difficult to be tested in humans. Here, we study some components of the serotonergic system during the pre-symptomatic phase in a well-characterized *Drosophila* PD model, *Pink1^B9^* mutant flies. We detected lower brain serotonin content in *Pink1^B9^* flies, accompanied by reduced activity of SERT before the onset of motor dysfunctions. We also explored the consequences of a brief early manipulation of the serotonergic system in the development of motor symptoms later in aged animals. Feeding young *Pink1^B9^* flies with fluoxetine, a SERT blocker, prevents the loss of dopaminergic neurons and ameliorates motor impairment observed in aged mutant flies. Surprisingly, the same pharmacological manipulation in young control flies results in aged animals exhibiting a PD-like phenotype. Our findings support that an early dysfunction in the serotonergic system precedes and contributes to the onset of the Parkinsonian phenotype in *Drosophila*.

## 1. Introduction

PD is a neurodegenerative disorder characterized by motor symptoms such as bradykinesia, tremor, and postural rigidity. These symptoms are evident when striatal dopamine (DA) content has decreased by at least 60–80%. This is associated with 50–60% degeneration of the dopaminergic neurons in the *Substantia Nigra pars compacta* (*SNpc*), which innervate the caudate-putamen nuclei (*striatum*), the major modulators of motor activity in vertebrates [1,2]. PD is also characterized by non-motor symptoms, including depression, anxiety, sleep disorders, and constipation, among others [3,4]. Some of the non-motor symptoms precede the appearance of motor symptoms, which has led to the definition of a pre-symptomatic or prodromal phase of PD characterized by non-motor symptoms followed by a symptomatic phase with the presence of both motor and non-motor features [3].

Although dopaminergic dysfunction is one of the main features in PD patients, other neural systems are also affected, including the serotonergic system [5,6]. Some of the alterations reported in the serotonergic system in patients with a PD diagnosis include reduced availability of the serotonin (5-HT) transporter (SERT) and the 5-HT receptor 1A in the striatum [7,8], reduction in striatal levels of both 5-HT and its metabolite 5-hydroxy indole acetic acid, and reduced expression of the rate-limiting enzyme in 5-HT biosynthesis, tryptophan hydroxylase (TPH) [9,10]. Interestingly, two recent studies have shown alterations in SERT availability in the caudate-putamen region prior to the establishment of dopaminergic pathology or the onset of motor symptoms in human carriers of mutations associated with genetic PD [11,12], suggesting that the serotonergic dysfunction precedes any dopaminergic alteration. Despite these findings, little is known about whether the serotonergic neural system contributes to the onset of motor symptoms or, in general, to the PD-associated changes in the long term.

At least two issues must be considered when thinking about this problem. First, serotonergic and dopaminergic systems are anatomical and functionally connected in mammals. Serotonergic projections from *Raffe Nuclei* innervate dopaminergic neurons from the *nigrostriatal* pathway, including both their somas in *SNpc* and their synaptic terminals in the *striatum* [13]. Moreover, 5-HT modulates the activity of dopaminergic neurons from *SNpc* [14,15]. Thus, it is possible that an alteration in the serotonergic system in the prodromal stage could contribute to the disturbance of the nigrostriatal dopaminergic circuit in PD. This is an important question that has not been thoroughly examined.

Secondly, it is a difficult task to study the state of the serotonergic system at the prodromal stages of PD. In this regard, animal models could play a key role in gaining better comprehension of the changes occurring at the pre-symptomatic stages of the disease. Several animal models for PD have helped us advance our understanding of the cellular and molecular mechanisms underlying the disease. One of the most studied *Drosophila* PD models is *Pink1^B9^*, generated by a deletion in the *Pink1* gene, which encodes the Ser/Thr kinase PINK1 (PTEN-induced putative kinase 1), linked to mitochondrial quality control [16,17,18,19]. We and others have reported that *Pink1^B9^* mutant flies exhibit loss of dopaminergic neurons correlated with motor impairment and also some non-motor phenotypes, such as olfactory defects, altered circadian rhythm, and impaired learning and memory [17,20,21]. We have previously proposed that in this PD animal model, there is a pre-symptomatic phase characterized by olfactory deficits followed by a symptomatic phase evidenced by locomotor defects, which start by the third-week post-eclosion [20]. This makes the *Pink1^B9^* mutant a suitable model to study the pre-symptomatic phase of PD.

Here, we have studied the serotonergic system in the pre-symptomatic phase of the *Pink1^B9^* model for PD and evaluated the long-term consequences of pharmacologically manipulating SERT activity early in the fly’s life. We observed lower 5-HT brain content and decreased SERT activity in *Pink1^B9^* mutants prior to the onset of motor defects. Moreover, we have demonstrated that pharmacological increase of serotonergic signaling by blocking SERT during the pre-symptomatic phase prevented the onset of the locomotor impairment and the loss of neurons in a particular dopaminergic neuronal population, the PPL2 cluster, in *Pink1^B9^* mutants. Remarkably, the same pharmacological manipulation caused PD-like locomotor defects and loss of PPL2 dopaminergic neurons in control flies. Our data support the notion that a disturbance in serotonergic signaling early in life contributes to dopaminergic neuronal degeneration and locomotor defects in older flies. This suggests that 5-HT is important for the maintenance of the dopaminergic system and its associated behaviors. Moreover, our results also suggest that reduced serotonergic signaling during the pre-symptomatic phase in *Pink1^B9^* mutant flies precedes and contributes to the onset of the later symptomatic phase.

## 2. Materials and Methods

### 2.1. Fly Stock and Maintenance

*Drosophila melanogaster* stocks were raised at 19 °C under a 12/12 h light/dark cycle (~50 flies per vial) and maintained on a standard yeast meal diet (10% yeast extract, 8% glucose, 5% flour, 1.1% agar, 0.6% propionic acid, and 1.2% nipagin). Only male flies were used in experiments. All flies used for experiments were collected from the moment of eclosion and maintained at 25 °C. *Pink1^B9^*, *SerT^MI02578^* and *w^1118^* (genetic control) flies were obtained from Bloomington *Drosophila* Stock Center (stock numbers #34749, #36004, and #5905, respectively). *SerT^MI02578^* mutant flies were isogenized to the *w^1118^* background for at least five generations.

### 2.2. Bioethical and Biosafety Issues

All experimental procedures were approved by the Bioethical and Biosafety Committee of the Pontificia Universidad Católica de Chile and were conducted in accordance with the guidelines of the Agencia Nacional de Investigación y Desarrollo (ANID) and the Servicio Agrícola y Ganadero de Chile (SAG).

### 2.3. Serotonin Measurement in Fly Brains by HPLC

5-HT measurements were carried out as we reported [21]. For each independent measurement, five adult *Drosophila* brains were dissected out and homogenized in 100 μL of 0.2 N perchloric acid by sonication. Afterward, the samples were passed through a Whatman PES 0.2 μm filter. Endogenous 5-HT tissue levels were quantitated by high-performance liquid chromatography (HPLC) coupled with electrochemical detection. For this, 5 μL of the sample was injected into an HPLC system (BAS, West Lafayette, IN, USA). The HPLC mobile phase, consisting of 0.1 M sodium phosphate monobasic, 1.8 mM 1-octane sulfonic acid, and 1 mM EDTA (pH 2.5), was pumped at a flow rate of 60 μL/min. The potential of the amperometry detector was set at 0.650 V. Under these experimental conditions, the retention time for 5-HT was 23.5 min. Samples were analyzed by comparing the peak area of 5-HT and its elution time with a reference standard for 5-HT.

### 2.4. Fast-Scan Cyclic Voltammetry (FSCV)

Ex vivo electrochemical recordings were performed as previously described [20,22]. A single fly brain was maintained in a recording chamber in the presence of a recording solution (NaCl 140 mM, KCl 4.5 mM, HEPES 10 mM, MgCl_2_ 1 mM, CaCl_2_ 2.5 mM, glucose 11 mM; pH 7.2) at 0.05 mL/min. A glass-carbon microelectrode was placed next to the fly brain to record responses after exposure to a bolus of 100 μM 5-HT. A triangle waveform was ramped from −0.4 to 1.2 V and back (vs. Ag/AgCl reference electrode) every 100 ms, with a scan rate of 400 V/s (Chem-Clamp potentiostat; Dagan Corporation, Minneapolis, MN, USA). Two National Instruments acquisition cards (NI-DAQ; PCI-6711 and PCI-6052e; National Instruments, Austin, TX, USA) were used to interface the potentiostat and stimulator with Demon Voltammetry and Analysis Software (Wale Forest Innovation, NC, USA).

### 2.5. Drug Exposure

Three-day-old flies were fed for three days of standard food supplemented with 15 μM or 150 nM fluoxetine hydrochloride (TOCRIS, Bristol, UK). The final concentration of the chemical was reached after diluting a 1 mM stock solution in molten fly food beforehand. Afterward, flies were transferred to vials containing normal food until they were 28 or 30 days old, when different analyses were performed. We used low doses of fluoxetine to avoid potential side effects, as described previously [23].

### 2.6. Video Recording of Motor Behavior

Fly’s behavior was monitored using a set-up we previously described [21,23,24]. Single flies were placed in a circular white arena (39 mm diameter, 2 mm high) with two pieces of cotton placed on opposite sides, and behavior was recorded for 3 min at room temperature (control condition). All video recordings were analyzed offline, using the tracking analysis software CeTrAn [25]. Data for different motor parameters were obtained; only distance traveled and activity time are presented here as the best descriptors of motor behavior in adult flies [20,22].

### 2.7. Immunofluorescence

Adult brains were dissected out and fixed with 4% paraformaldehyde for 20 min. After that, samples were rinsed with PBT (PBS 1× +0.3% Triton X-100) and then blocked with BSA 1% in PBT + sodium azide 0.03% for 20 min. Tissue was incubated with sheep polyclonal anti-TPH/Trh antibody (AB1541, Millipore, Burlington, MA, USA) (1:50) for three days at 4 °C to identify serotonergic neurons or with rabbit polyclonal anti-TH antibody (AB152, Millipore, Burlington, MA, USA) (1:300) for one day at 4 °C to identify dopaminergic neurons. The use of the anti-TPH antibody in immunohistochemistry was previously validated by Bao and colleagues [24]. Then, samples were rinsed with PBT and incubated for 2 h at room temperature with secondary Alexa Fluor 568 donkey anti-sheep (A-21099, Thermo Fisher Scientific, Waltham, MA, USA) (1:500) or Alexa Fluor 568 goat anti-rabbit (A-11011, Thermo Fisher Scientific, Waltham, MA, USA) (1:500) to visualize serotonergic or dopaminergic neurons, respectively. Finally, brains were rinsed with PBT and mounted using SlowFade™ Gold Antifade Mountant with DAPI (S36938, Thermo Fisher Scientific, Waltham, MA, USA).

### 2.8. Imaging and Quantification of Trh and TH-Positive Cells

Whole-mount adult brains stained for Trh or TH were imaged at a confocal microscope Nikon C2 with a spectral detector (Nikon Instruments Inc., Melville, NY, USA), at the Advanced Microscopy Facility UMA BIO UC, Pontificia Universidad Católica de Chile. Images were taken using the objective Plan Apo VC 20× DIC N2 NA 0.75. High-resolution images (1024 × 1024 pixels) were obtained using the NIS-Elements C Nikon software (Nikon Instruments Inc., Melville, NY, USA) and processed in ImageJ 1.52p software (National Institutes of Health, Bethesda, MD, USA). A reconstruction of the whole brain was performed, and positive cells were counted manually in each brain stack. Ten and fifteen brains were used per condition to count the number of serotonergic and dopaminergic neurons, respectively. Left and right brain hemispheres were counted independently. ADMP, AMP, ALP, PMPD, PMPM, PLP, PMPV, LP, SEL, and SEM serotonergic clusters and PPL1 and PPL2 dopaminergic clusters were identified according to their anatomical position, as described in the literature [20,26,27,28]. Two different people, blind concerning genotype or experimental condition, carried out quantifications.

### 2.9. Western Blot

Samples were obtained from 60 heads of *w^1118^* control and *Pink1^B9^* mutant adult male flies. Homogenized samples (30 μg of total proteins) were heated at 95 °C for 5–10 min and then separated by SDS-PAGE. Proteins were transferred to a nitrocellulose membrane, which was incubated with PBS 1× 0.1% Tween-20, and 3% BSA for 1 h at room temperature. Membranes were then incubated with sheep anti-Trh antibody (AB1541, Millipore, Burlington, MA, USA) (1:1000) and with mouse anti-beta-actin antibody (ab8224, Abcam, Cambridge, UK) (1:6000) overnight at 4 °C. Afterward, membranes were rinsed twice with PBS 1× and twice with PBST (PBS 1× + 0.1% Tween—20), followed by incubation with donkey anti-sheep IgG antibody (A16041, Thermo Fisher Scientific, Waltham, MA, USA) (1:3000) or goat anti-mouse IgG antibody (62–6520, Thermo Fisher Scientific, Waltham, MA, USA) (1:5000) for 2 h at room temperature. Then, membranes were washed with PBS 1× and PBST. Proteins were visualized by chemiluminescence (ECL, Amersham, Biosciences, Piscataway, NJ, USA).

### 2.10. Real-Time Quantitative PCR

RNA was obtained from about 50–100 adult heads by using TRIzol™ Reagent (15596018, Thermo Fisher Scientific, Waltham, MA, USA). Reverse transcription was carried out with 3 μg of RNA (K1622, Revertaid First Strand cDNA Synthesis kit, Thermo Fisher Scientific, Waltham, MA, USA) according to the manufacturer’s instructions. Afterward, 150 ng of cDNA was amplified with 5X HOT FIREPol EvaGreen qPCR Mix Plus (08-24-00001, Solis Biodyne, Tartu, Estonia), in the presence of specific primers, in the Light Cycler 480 system (Roche, Basel, Switzerland). The PCR program consisted of 15 min at 95 °C followed by 35 cycles of 15 s at 95 °C, 20 s at the annealing specific temperature (for each pair of primers), and 30 s at 72 °C for 35 cycles. Quantification of PCR products was performed using the Pfaffl method [29]. PCR reactions were carried out in duplicate in three or more independent biological replicas. Data for *Pink1* and *SerT* expression was normalized to *Gapdh2* expression (housekeeping gene). The following primers were used to amplify *Pink1*: F Pink1 5′—TCGGTGGTCAATGTAGTGCC—3′ and R Pink1 5′—TCGGTGGTCAATGTAGTGCC—3′. PCR product was 200 bp, and the efficiency of primers was 1.85. To amplify *SerT*, the following primers were used: F SERT 5′—CAACAACGAGCGCATTCTG—3′ and R SERT 5′—GAAGATGAGGAAGAGGCAGTAG—3′. PCR product was 200 pb, and the efficiency of primers was 2.05. For amplifying *Gapdh2*, we used: F Gapdh2 5′—CGTTCATGCCACCACCGCTA—3′ and R Gapdh2 5′—CCACGTCCATCACGCCACAA—3′. The PCR product was 72 bp, and the efficiency of primers was 2.24. Conditions for PCR reaction of *SerT* were as indicated above, but reactions were carried out with 600 ng of RNA as a template for reverse transcription.

### 2.11. Statistical Analysis

Statistical analysis was performed using GraphPad Prism 7 (GraphPad Software, San Diego, CA, USA) and R Studio using the *rcompanion*, *DescTools*, and *rstatix* packages. All data throughout the text are shown as mean ± SD. The Shapiro–Wilk test for normality was used to determine whether parametric or non-parametric statistical tests were to be used. A Student t-test was performed to compare two groups of datasets. Kruskal–Wallis’s non-parametric test was used when comparing data and one factor. A two-way ANOVA test with Bonferroni’s analysis as a post hoc test or Scheirer-Ray-Hare non-parametric test with Dunn’s analysis as a post hoc test was used to evaluate multiple groups with two independent factors. The criteria for significance were ns (not significant), * *p* < 0.05, ** *p* < 0.01, *** *p* < 0.001, **** *p* < 0.0001. The number of animals tested (“*n*”) and the statistical analysis for each experiment dataset are indicated in figure legends.

## 3. Results

### 3.1. Pink1^B9^ Mutant Flies Exhibit Lower 5-HT Brain Content during the Pre-Symptomatic Phase

The *Pink1^B9^* mutant fly was generated by deletion of 570 bp in the endogenous *Drosophila Pink1* gene located in chromosome X, resulting in no detectable levels of *Pink1* transcripts (Appendix A) [17]. We have previously shown that hemizygous male *Pink1^B9^* mutants begin to exhibit locomotor deficits by the third-week post-eclosion, supporting the notion of pre-symptomatic and symptomatic stages characterized by the absence and presence of locomotor defects, respectively [20].

To begin examining the status of the serotonergic system in this PD animal model, we measured the brain content of 5-HT in *Pink1^B9^* hemizygous mutants and *w^1118^* male flies throughout the adult stage. Our data show similar 5-HT levels in mutant and control flies during the first-time window analyzed (0–3 days post-eclosion) (Figure 1). However, as flies age, the brain 5-HT content shows genotype-specific changes (Figure 1, H(1,60) = 8.047; *p* = 0.0046). *Pink1^B9^* mutants exhibit lower 5-HT content by 7–10 (92.9 ± 26.6 fmol/brain) and 14–17 days post-eclosion (60.7 ± 12.8 fmol/brain) as compared to 7–10 (178.2 ± 45.1 fmol/brain; *p* = 0.0202) and 14–17 days-old (166.4 ± 87.1 fmol/brain; *p* = 0.0087) control flies. These differences between genotypes are no longer observed by 21–24 and 28–31 days post-eclosion. Surprisingly, the interaction of the genotype and time contributes to the differences in 5-HT content (H(4,60) = 15.724; *p* = 0.0034), suggesting that the content of this amine varies differently in control and mutant flies as they age. Together, these results show that in the *Pink1^B9^* mutant, the pre-symptomatic phase is characterized by reduced brain 5-HT content, which precedes the onset of the symptomatic phase at later time windows.

To advance on the mechanisms underlying the reduced brain 5-HT content in *Pink1^B9^* mutants, we evaluated the status of some components of the serotonergic system in control and *Pink1^B9^* flies. First, we quantified the number of serotonergic neurons. We aimed to determine whether a reduced number of serotonergic neurons explains the lower brain 5-HT content in *Pink1^B9^* mutants. To identify serotonergic neurons, we used an anti-TPH antibody that recognizes the *Drosophila* TPH, the rate-limiting enzyme for 5-HT biosynthesis (identified as Trh in flies). We analyzed the brains from 14-day-old flies from each genotype since this is the last time window we have studied in the pre-symptomatic phase in the *Pink1^B9^* fly PD model [20]. Our data showed no differences in the number of Trh positive cells in 14-day-old *Pink1^B9^* mutants compared to control flies (Appendix A), either by analyzing the number of neurons per serotonergic cluster (Appendix A; F(1,174) = 0.1230, *p* = 0.7262) or the total number of aminergic cells in the whole fly brain (Appendix A; t(18) = 0.2740, *p* = 0.7872). We also assessed the expression of Trh in whole head lysates by Western blot to determine whether a decrease in Trh protein expression explains the reduced brain 5-HT content in mutant flies. No differences were observed in Trh protein levels in 14–17-day-old *Pink1^B9^* mutants compared to age-matched control flies (Appendix A; t(8) = 1.074, *p* = 0.3143). Altogether, these findings indicate that the lower brain 5-HT content in *Pink1^B9^* mutants is not associated with lower expression of the Trh enzyme or a reduction in the number of serotonergic neurons.

### 3.2. SERT Activity Declines in Pink1^B9^ Mutant Flies during the Pre-Symptomatic Phase

As previously indicated, different studies have reported changes in the availability of SERT in the caudate-putamen in patients with a diagnosis of PD (that is, at symptomatic stages) [7,8,9,10]. Interestingly, alterations in SERT availability have also been reported in people that are carriers of mutations associated with this disease but who show no motor dysfunction (who could be considered as going through a pre-symptomatic stage) [11,12]. As explained above, the *Pink1^B9^* mutant strain seems to recapitulate most of the changes occurring through PD progression. We decided to evaluate the expression and activity of SERT during the pre-symptomatic phase in *Pink1^B9^* mutants in order to assess changes in this transporter. In the absence of an anti-SERT antibody that we could freely use, we assessed the *SerT* transcripts expression. Analysis of *SerT* transcript levels showed no differences between mutant and control animals when they were 3 days old (Appendix A; t(8) = 0.5502, *p* = 0.5972) or 14 days old (Figure 2a; t(8); 0.1801, *p* = 0.8615).

To study SERT activity, we evaluated the reuptake of exogenously applied 5-HT in brains obtained from *Pink1^B9^* mutant and control animals by in vitro FSCV. We studied two parameters that help assess the operation of the transporter: the half-life of the electrochemical signal detected when the brains are exposed to the amine and *tau*, a constant that reflects the kinetics of 5-HT reuptake. The results show a significant increase in *tau* (Figure 2b,c; t(34) = 3.293, *p* = 0.0023) in the brains of 14-day-old *Pink1^B9^* mutants (Figure 2b,c; 37.95 ± 10.80) when compared to age-matched control flies (Figure 2b,c; 27.81 ± 7.335). A significant increase in half-life (Figure 2b,d; t(34) = 3.293, *p* = 0.0023) was also detected in brains of 14-day-old mutants (Figure 2b,d; 26.18 ± 7.453 s) compared to control flies (Figure 2b,d; 19.19 ± 5.061 s). The same experiment was performed using younger flies (3 days old) but no differences were detected either in *tau* (Appendix A; t(33) = 0.07558, *p* = 0.9402) or in half-life (Appendix A; t(33) = 0.01852, *p* = 0.9853) when comparing control and mutant animals. Altogether, these data indicate that during later stages of the pre-symptomatic phase in *Pink1^B9^* mutant flies, there is a decline in SERT activity, with no changes in its expression.

### 3.3. A Transient Increase in Serotonergic Signaling in Young Control and Pink1^B9^ Mutant Flies: Opposite Effects on Locomotion

*Pink1^B9^* mutants exhibit lower 5-HT content when they are 7–10 and 14–17 days old compared to age-matched control flies (Figure 1). A reduced 5-HT content might result in decreased serotonergic signaling in mutant animals. Thus, we hypothesized that a manipulation intended to restore serotonergic signaling in young *Pink1^B9^* mutants could affect the timing for the onset of locomotor defects in this PD model. To test this idea, we decided to feed young *Pink1^B9^* mutants fluoxetine (15 µM), a SERT selective blocker, for three days (from day 3 to day 6 post-eclosion), i.e., right before the differences in 5-HT levels between mutant and control flies are evidenced. After the three days of treatment, flies were maintained in normal fly food until flies were 28 days old, when locomotion assays were carried out (Figure 3a). Two motor parameters were recorded, distance traveled (Figure 3b; H(1,88) = 24.28; *p* < 0.0001) and activity time (Figure 3c; H(1,88) = 21.21; *p* < 0.0001).

Our data show a decrease in the two motor parameters in untreated 28-day-old *Pink1^B9^* flies (Figure 3b,c; distance traveled = 62.091 ± 61.288 mm, *p* < 0.0001; activity time = 11.865 ± 12.914 s, *p* < 0.0001) compared to control flies (distance traveled = 323.326 ± 236.209; activity time = 51.894 ± 32.409 s), which is consistent with our previous report [20]. Treating young *Pink1^B9^* mutants with 15 µM fluoxetine significantly improved locomotion, with an increase in distance traveled (Figure 3b; 187.983 ± 149.441 mm, *p* = 0.0087) and activity time (Figure 3c; 34.925 ± 25.632 s, *p* = 0.00630) compared to untreated mutants (Figure 3b,c; distance traveled = 62.091 ± 61.288 mm; activity time = 11.865 ± 12.914 s). Interestingly, motor parameters recorded in fluoxetine-treated *Pink1^B9^* mutant flies are affected so they become more similar in magnitude to those observed in untreated control flies. This seems to be a dose-dependent effect since at a lower concentration of the SERT blocker (150 nM), this response is no longer observed (Appendix A; distance traveled H(1,109) = 3.565, *p* = 0.0590; activity time H(1,109) = 11.652, *p* = 0.2804).

Surprisingly, the same manipulation had the opposite effect on control flies. Fluoxetine-fed control flies displayed a decrease in distance traveled (Figure 3b; 110.961 ± 17.915 mm, *p* = 0.0009) and activity time (Figure 3c; 23.449 ± 19.241 s, *p* = 0.00725) compared to untreated control flies (distance traveled = 323.326 ± 236.209 mm; activity time = 51.894 ± 32.409 s). This seems to be also a dose-dependent effect (Appendix A) since no effect is observed when control flies are treated with 150 nM fluoxetine.

To confirm the specificity of the fluoxetine effect on motor activity, we attempted a genetic manipulation aimed at the same molecular target, *SerT*. We used a *SerT* hypomorphic *Drosophila* mutant that we previously demonstrated exhibits reduced transporter expression [22]. Reduced *SerT* expression results in decreased transporter activity in the membrane, which leads to increased extracellular 5-HT levels and, therefore, increased serotonergic signaling (Appendix A). Our results show that the genetic reduction in *SerT* expression in the *Pink1^B9^* mutant background (*Pink1^B9^,SerT^MI02578^* double mutant animals) prevented the onset of motor defects in 28-day-old flies, i.e., *Pink1^B9^,SerT^MI02578^* flies exhibit an increase in distance traveled (1430 ± 1012 mm) and activity time (112.8 ± 55.65 s) compared to *Pink1^B9^* mutant flies (Appendix A; distance traveled = 414.4 ± 659.0 mm, *p* = 0.0009; activity time = 37.10 ± 36.03 s, *p* = 0.0001).

Altogether, these results indicate that an increase in serotonergic signaling early in life in *Pink1^B9^* mutants prevents—at least in part—the locomotor defects associated with this animal model of PD. Interestingly, when this treatment is carried out on control flies, it evokes the onset of motor defects as animals age.

### 3.4. A Transient Increase in Serotonergic Signaling in Young Control and Pink1^B9^ Mutant Flies: Opposite Effects on Survival of PPL2 Dopaminergic Neurons

We previously described dopaminergic cell loss in specific neuronal clusters in *Pink1^B9^* mutants, including PPL1 and PPL2 populations, as flies age and the locomotor defects become evident [20]. We proposed to evaluate whether the transient fluoxetine treatment, which was efficient at improving motor behavior in aged mutant flies, has any effect on the loss of dopaminergic neurons. Thus, we quantified the number of PPL1 (Figure 4c; H(1,55) = 2.925; *p* = 0.0872) and PPL2 (Figure 4d; H(1,56) = 20.713; *p* < 0.0001) dopaminergic neurons in 30-day-old flies fed fluoxetine (15 µM), as compared to non-treated animals (Flx. and Ctrl conditions, respectively; Figure 4a–d). The number of PPL1 (Figure 4c; 10.933 ± 0.961 neurons per hemisphere) and PPL2 (Figure 4d; 4.600 ± 0.828 neurons per hemisphere) neurons in 30-day-old flies were reduced in *Pink1^B9^* mutant flies compared to control flies (Figure 4c,d; PPL1 = 12.429 ± 0.756 neurons per hemisphere and PPL2 = 5.667 ± 0.900 neurons per hemisphere), consistent with what we and others have previously reported [17,20]. Interestingly, while fluoxetine treatment did not affect the number of PPL1 neurons in *Pink1^B9^* mutants (Figure 4c; Ctrl. = 10.933 ± 0.961 vs. Flx. = 11.667 ± 1.047 neurons per hemisphere, *p* = 0.3984), it induced an increased number of PPL2 neurons in fluoxetine-treated *Pink1^B9^* mutants compared to untreated mutant flies (Figure 4d; Ctrl. = 4.600 ± 0.828 vs. Flx. = 5.667 ± 1.345 neurons per hemisphere, *p* = 0.0063). Remarkably, this is similar to the number of neurons recorded in untreated control flies (Figure 4d; Ctrl. = 5.667 ± 0.900 neurons per hemisphere). This suggests a protective effect of fluoxetine on this dopaminergic cluster.

Surprisingly, a lower number of PPL2 neurons was observed in control flies treated with fluoxetine compared to control untreated animals (Figure 4d; Ctrl. = 5.667 ± 0.900 vs. Flx. = 4.000 ± 0.845 neurons per hemisphere, *p* = 0.0073), while no effect in the number of PPL1 neurons was detected in this condition (Figure 4c; Ctrl. = 12.429 ± 0.756 vs. Flx. = 12.267 ± 0.884 neurons per hemisphere, *p* > 0.9999).

Together, these results suggest a role of the fluoxetine treatment on the survival of PPL2 dopaminergic neurons: while the transient increase in serotonergic signaling in young control flies is toxic and promotes the loss of PPL2 neurons as these animals age, the same pharmacological manipulation in young *Pink1^B9^* mutants is beneficial, which is evidenced by prevention of the loss of PPL2 neurons in this PD animal model.

## 4. Discussion

Alterations in components of the serotonergic system have been described in the early stages of the symptomatic phase in patients with PD and also in postmortem samples [7,8,9,10,30]. Reports on a reduction in 5-HT content in caudate and putamen in the postmortem samples of PD patients [9] are consistent with the idea that as the disease progresses, not only the dopaminergic but also the serotonergic system is affected. It is known that an important proportion of PD patients experience depression and mood alterations as the disease evolves, and they are regularly prescribed antidepressants (usually serotonergic agents) in addition to dopaminergic therapy [31,32,33].

It is a particularly difficult task to assess whether the serotonergic system contributes to the progression of motor and non-motor features in this disease, especially at the early stages [6]. However, two recent works have provided support to this hypothesis. In 2017, a PET study performed in human carriers of a mutation in the gene LRRK2 associated with PD demonstrated, for the first time, an alteration in SERT binding in the *striatum* when compared to control subjects before the onset of motor symptoms [11]. In 2019, a different PET work showed that a reduction in SERT binding in the *striatum* precedes changes in the dopaminergic system and the onset of motor deficits in human carriers of the *A53T* mutation in the α-synuclein gene, which is linked to an autosomal-dominant form of PD [12]. These two studies show changes in SERT binding that precede the establishment of the dopaminergic pathology and motor symptoms in PD. Similar to the studies carried out in humans, we also found alterations in components of the serotonergic system in the pre-symptomatic phase in the *Pink1^B9^ Drosophila* PD model. In our study, we were able to perform pharmacological and genetic manipulations of the serotonergic system early in the life of flies and study the consequences of these treatments later in older animals. These experiments showed that these manipulations early in young animals may be beneficial for a fly with a genetic predisposition to develop parkinsonian phenotypes but harmful for a control fly.

Our results show reduced brain 5-HT content in *Pink1^B9^* mutant brains during the late time windows of the pre-symptomatic phase, as compared to control flies. This could be explained by the reduced activity of SERT: decreased 5-HT reuptake leads to minor recycling of the amine back into the pre-synaptic nerve terminal and, therefore, a reduced tissue content of this monoamine. However, the deficiency in SERT activity in *Pink1^B9^* mutant brains could also be viewed as a compensatory mechanism in response to the lower serotonergic signaling, i.e., by decreasing the reuptake of 5-HT, it is possible to maintain the amine concentration in the extracellular milieu at the required level to activate its receptors and prolong its actions on target neurons. Interestingly, the reduced activity of SERT in this *Drosophila* PD model is reminiscent of the PET study of Wilson et al. (2009). In that study, authors reported decreased binding of a SERT radiotracer in the caudate-putamen region in human carriers of the *A53T SNCA* gene mutation when they still show no signs of motor alterations. Since we did not detect differences in *SerT* transcripts expression, the reduced transporter activity in *Pink1^B9^* flies may be the result of the internalization of SERT or the alteration in its trafficking to the cell surface, resulting in a decrease in 5-HT reuptake. Although these are ideas that have not been evaluated for SERT, it has been recently discussed how PD-linked mutations could interrupt normal DAT trafficking, leading to alterations in DA homeostasis [34]. Whether SERT trafficking and membrane localization are also affected by PD-associated mutations is a relevant question that deserves further studies.

Our results, demonstrating the beneficial effects of an increase in serotonergic signaling in flies carrying out a mutation associated with an animal model of PD, are consistent with a previous work carried out in MPTP-treated monkeys, where it was shown that motor behavior recovery was associated with increased 5-HT levels in the *striatum* [35]. A different work showed that fluoxetine treatment after i.p. injections of MPTP prevented the degeneration of *nigrostriatal* dopaminergic neurons, with partial motor recovery [36]. Since the SERT blocker increases the extracellular levels of 5-HT in the brain, it is possible to propose that increased serotonergic signaling is responsible for the neuroprotective effect observed in these works.

As previously mentioned, serotonergic circuits modulate the activity of dopaminergic pathways in the mammalian brain [13,14,15]. Although the connection between a specific pair of serotonergic neurons and three PPL1 neurons in the mushroom body peduncle [37] has been demonstrated, whether serotonergic neurons innervate and modulate the activity of dopaminergic cells in the *Drosophila* brain is an issue that has not been thoroughly studied. Connectomic data from electron microscopy studies available at *Virtual Fly Brain* (https://www.virtualflybrain.org, accessed on 19 April 2022) show that PPL1 and PPL2 neurons are innervated by serotonergic fibers. Furthermore, *Drosophila* single-cell transcriptomic data, available at *Fly Atlas Cell* (https://www.flycellatlas.org, accessed on 19 April 2022), identify the expression of 5-HT receptors in several dopaminergic neurons expressing TH, which could potentially include PPL1 and PPL2 neurons. Therefore, treatments targeting serotonergic transmission might also affect dopaminergic neurons innervated by 5-HT fibers. Serotonin could be activating its serotonergic receptors on dopaminergic neurons to trigger a trophic-like action (see also below). Fanibunda et al. (2019) reported that mitochondrial biogenesis is regulated by the 5-HT2a receptor and SERT-PGC-1α in rodent cortical neurons [38]. Since the *Pink1^B9^* mutant fly is a PD model exhibiting mitochondrial dysfunction, it is possible that the neuroprotective effect of fluoxetine by 5-HT could come from the regulation of mitochondrial biogenesis and function in dopaminergic neurons, particularly in PPL2 neurons. More experiments are needed to test if this is the case.

Regarding the protection of dopaminergic neurons, we detected that two different dopaminergic clusters that degenerate in *Pink1^B9^* mutant flies are differentially affected by fluoxetine treatment. We suggest that the effect of fluoxetine is more effective in the PPL2 cluster in *Pink1^B9^* mutants as the treatment was carried out from day 3 until day 6 post-eclosion. By that time, the number of PPL1 neurons is already decreased, as previously shown [20]. On the other hand, PPL2 is a cluster that degenerates as the pathology progresses in *Pink1^B9^* mutants [20]. Thus, the treatment with fluoxetine did not protect PPL1 neurons due to the earlier commitment of this population to cell death. Interestingly, this differential effect suggests that the treatment is preventive rather than restorative.

It has been proposed that 5-HT may act as a trophic factor as it participates in the establishment of CNS neuronal networks in the mammalian brain [39,40]. In the adult stage, brain 5-HT homeostasis is also essential for the maintenance of serotonergic circuits. Actually, it has been shown that the perturbation in 5-HT homeostasis during the adult brain affects serotonergic fiber density in mice [41]. On the other hand, Simpson et al. (2011) showed that a perinatal treatment with citalopram, a SERT blocker, results in diverse structural and electrophysiological abnormalities in rats [42]. These findings are similar to our demonstration of the deleterious effect of fluoxetine treatment on control flies.

Overall, our data suggest that there are dynamic changes in 5-HT levels occurring in the *Drosophila* brain to maintain an adequate level of serotonergic signaling. Considering the evidence of a trophic effect of 5-HT in mammalian neural circuits, we propose that this amine plays such a role in participating in the maturation and refinement of specific brain circuits in the fly during a time window called the critical period. The existence of such a critical period has been previously discussed, when *Drosophila* neural circuits undergo refinement, similar to what is described in the vertebrate brain [43]. This time window would span the first week after flies eclose. For instance, it has been reported that the *Drosophila* brain circuits that are responsible for olfactory learning and memory are refined in adult flies over the first week of age [44,45,46]. Here, we show that hampering normal 5-HT dynamic changes by the fluoxetine treatment in this critical period in control individuals leads to the death of PPL2 dopaminergic neurons and impaired locomotion. Conversely, the same treatment was beneficial for the fly PD model, i.e., fluoxetine feeding in young *Pink1^B9^* mutant flies prevented the onset of locomotor defects and the loss of PPL2 dopaminergic neurons.

Our results and the literature support the working hypothesis that there is a range of 5-HT concentrations that is protective for the fly brain (Figure 5). Thus, young control flies reach the 5-HT protective levels (blue line, gray zone), promoting the normal maturation of neuronal circuits and survival of dopaminergic neurons, which would underlie normal motor output. In *Pink1^B9^* mutants, the 5-HT levels are lower and out of the protective range (red line, white zone), hindering the normal maturation of circuits and, therefore, leading—later in life—to the symptomatic phase of this disease model. We propose that after the pharmacological treatment with fluoxetine, 5-HT levels in *Pink1^B9^* mutant flies achieve the protective range (dashed red line, gray zone), preventing the beginning of the Parkinsonian phenotype. On the contrary, after the fluoxetine treatment, 5-HT levels in control flies go out of the protective range (dashed blue line, white zone), hampering the normal aminergic signaling and carrying consequences on brain maturation and the expression of behaviors in aged animals, such as the Parkinsonian phenotype.

In sum, we have provided new data supporting the idea that a disturbance in serotonergic signaling early in life contributes to dopaminergic neuronal degeneration and locomotor defects in older flies, suggesting that 5-HT could be fundamental for the maintenance of the dopaminergic system and its associated behaviors. Remarkably, our findings in the PD fly model agree with recent studies in humans [11,12], supporting that an early commitment of the serotonergic system—particularly SERT—is a general pathological hallmark for PD. New approaches and tools are becoming available for an early diagnosis of PD, even at pre-symptomatic stages [33,47,48,49,50]. Therefore, the clinical practice needs to find new tools for preventing or delaying the onset of PD symptoms. Our data suggest that increasing serotonergic signaling in the pre-symptomatic stage of this disease would serve this purpose. Some of these new tools could be based on SERT, the molecular target of fluoxetine.

## Figures and Tables

**Figure 1 cells-11-01544-f001:**
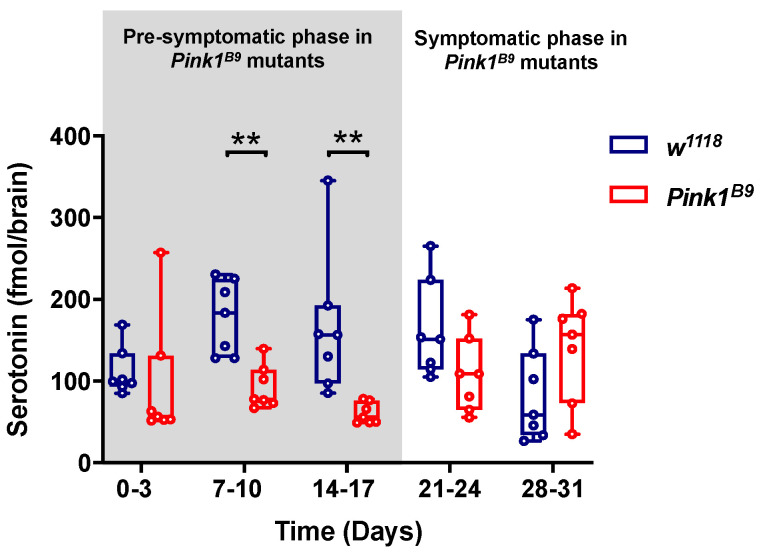
*Pink1^B9^* mutants exhibits lower 5-HT brain content during the pre-symptomatic phase. Serotonin content in brains of *Pink1^B9^* mutants and *w^1118^* control flies throughout adult life, measured by HPLC coupled to an electrochemical detector. Seven independent measures per time window are shown; each measurement corresponds to 5 brains (35 brains in total per genotype). Data represent the interquartile range with maximum and minimum ranges. ** indicates *p* < 0.01, when comparing genotypes. Scheirer-Ray-Hare test followed by Dunn’s post hoc test. Pre-symptomatic (gray background) and symptomatic phases of *Pink1^B9^* mutant flies are indicated.

**Figure 2 cells-11-01544-f002:**
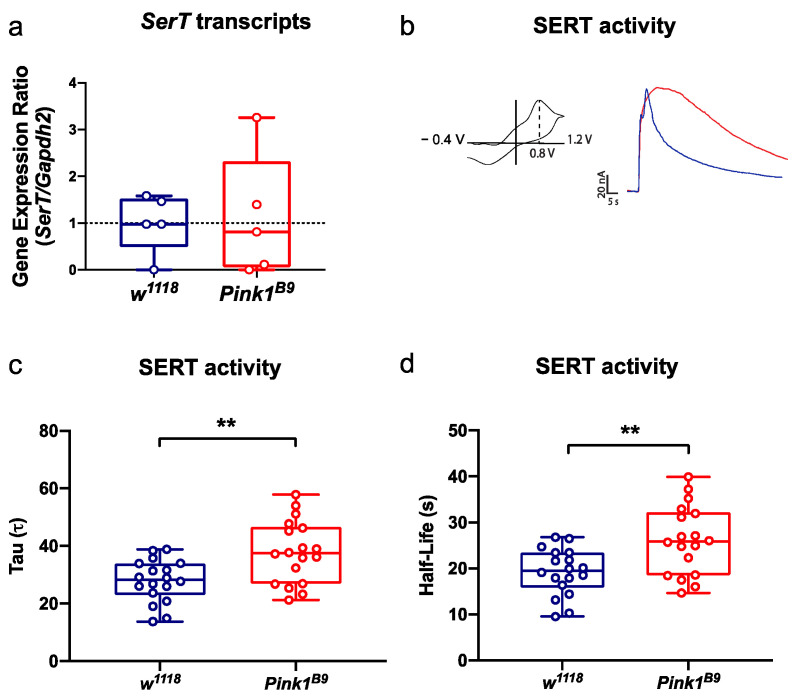
The activity of SERT declines during the pre-symptomatic stage in *Pink1^B9^* mutant flies. (**a**) *SerT* transcript levels were evaluated in 14-day-old *w^1118^* control and *Pink1^B9^* mutant flies by RT-qPCR. Quantification is presented as the gene expression ratio of transcripts detected for *SerT* normalized to *Gapdh2* transcripts. Data are presented as the interquartile range with maximum and minimum ranges. Student’s t-test indicates no statistical differences between groups. (**b**–**d**) SERT activity in *Pink1^B9^* mutant and *w^1118^* control animals evaluated by FSCV in brains of 14-day-old flies. (**b**) In the left panel is shown a typical voltammogram for 5-HT, while representative experiments showing 5-HT signals recorded in brains of *w^1118^* control (blue line) and *Pink1^B9^* mutant flies (red line) are shown in the right panel. (**c**) *tau* and (**d**) half-life kinetic parameters obtained from electrochemical recordings in *w^1118^* control and *Pink1^B9^* fly brains exposed to 5-HT reflect the operation of SERT. An increase in these parameters reflects a less active uptake of the amine. Data in (**c**,**d**) are presented as the interquartile range with maximum and minimum ranges of recordings from 5 brains per genotype/condition. ** indicates *p* < 0.01; Student’s *t*-test.

**Figure 3 cells-11-01544-f003:**
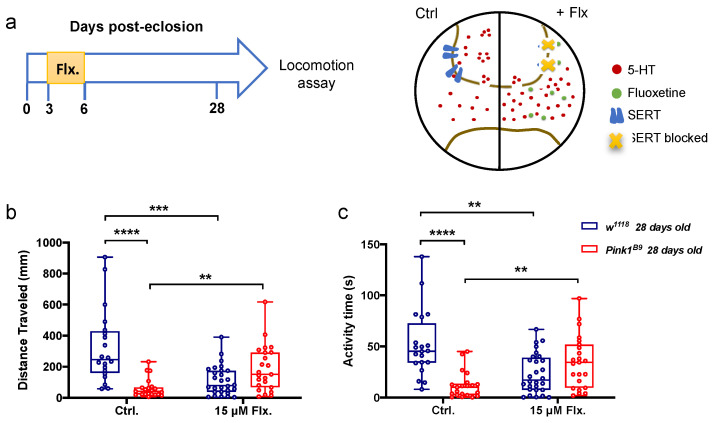
Effect of fluoxetine treatment on the locomotor behavior in 28-day-old *Pink1^B9^* mutant and control flies. (**a**). Left panel, a scheme for the timing of the fluoxetine (15 µM) treatment. Right panel, a schematic representation of the pharmacological effect of fluoxetine treatment (right, +Flx) compared to the control situation (left, Ctrl). (**b**,**c**) Distance traveled (mm) and activity time (s), two locomotor parameters recorded in 28-day-old control and *Pink1^B9^* mutant flies treated or not with fluoxetine (15 µM Flx and Ctrl, respectively). Data was obtained from *n* = 20 (*w^1118^*); 28 (*w^1118^* + Flx); 21 (*Pink1^B9^*); 23 (*Pink1^B9^* + Flx) flies. Data are presented as the interquartile range with maximum and minimum ranges. **, *** and **** indicate *p* < 0.01, *p* < 0.001 and *p* < 0.0001, respectively; Scheirer-Ray-Hare test followed by Dunn’s post hoc test.

**Figure 4 cells-11-01544-f004:**
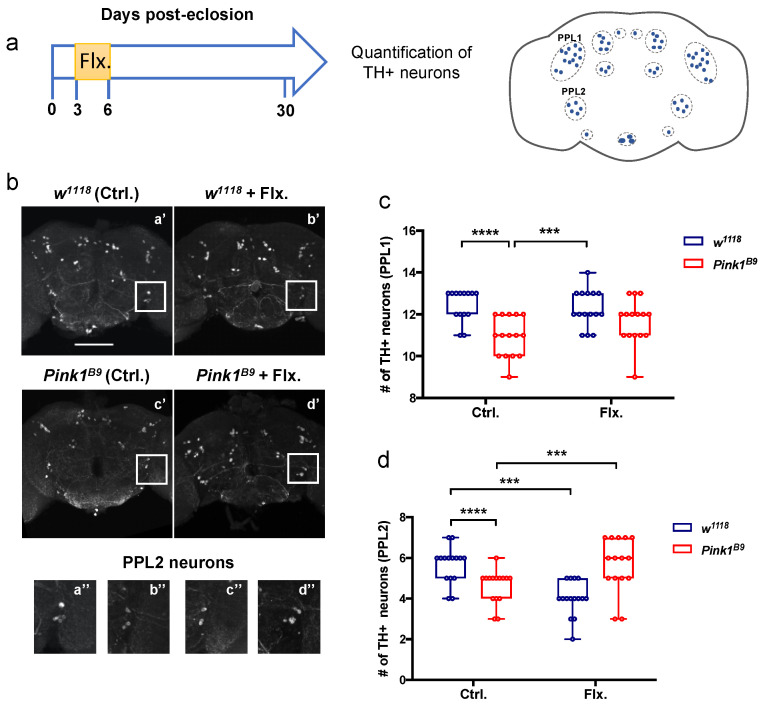
Effect of fluoxetine treatment on the survival of dopaminergic neurons in *Pink1^B9^* mutant and control flies. (**a**). Left panel, a scheme for the timing of fluoxetine (15 µM) treatment. Right panel, representation of the TH-positive clusters identified in the posterior portion of the adult fly brain. (**b**) Representative images of posterior fly brain regions in 30-day-old *w^1118^* and *Pink1^B9^* flies after immunofluorescent staining with an anti-TH antibody. Flies were treated (+Flx, (**b’**,**d’**)) or not (**a’**,**c’**) with fluoxetine. Magnification of area indicated in (**a’**–**d’**) is shown in (**a”**–**d”**). Scale bar: 100 µm. (**c**,**d**) Quantification of PPL1 and PPL2 TH-positive neurons in each genotype and experimental condition. Data are presented as the interquartile range with maximum and minimum ranges of quantifications in 15 brains per condition. *** and **** indicate *p* < 0.001 and *p* < 0.0001, respectively; Scheirer-Ray-Hare test followed by Dunn’s post-test.

**Figure 5 cells-11-01544-f005:**
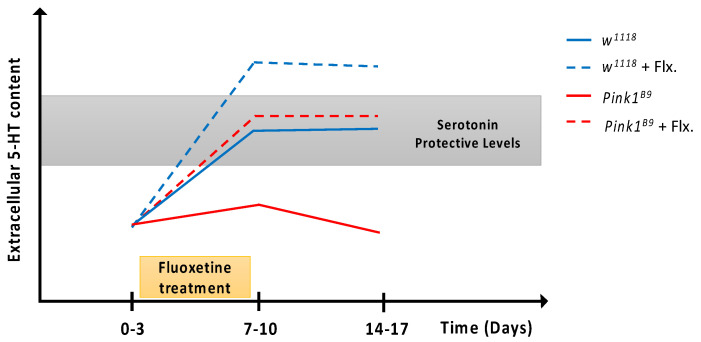
A proposal for the protective vs. toxic effects of serotonin in the fly brain. Schematic representation of the dynamic changes in extracellular 5-HT content proposed to be occurring in the adult fly brain under the different conditions studied. Control flies exhibit a protective/trophic level of 5-HT content (blue line, gray zone), which allows the normal maturation and refinement of neuronal circuits that participate in motor behavior. In *Pink1^B9^* flies, no protective/trophic 5-HT levels are reached (red line, white zone), hindering the normal maturation and refinement processes, which favors the onset of Parkinsonian phenotypes in aged animals. A transient increase (fluoxetine treatment, yellow box) of extracellular 5-HT levels in mutant flies serves to reach the protective range (red dashed line), permitting the normal maturation and refinement of neuronal circuits and halting the onset of the PD symptomatic phase later in life. On the other hand, in control flies, the fluoxetine treatment (yellow box) results in extracellular 5-HT levels that increase over the normal range (blue dashed line, white zone), disturbing the normal maturation and refinement of circuits, leading to the onset of PD-like phenotypes as animals age.

## Data Availability

All data are deposited at Figshare (https://figshare.com/s/8a82596ebcb37cfcdf6b, accessed on 19 April 2022).

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
