# Peer review of "An Early Disturbance in Serotonergic Neurotransmission Contributes to the Onset of Parkinsonian Phenotypes in Drosophila melanogaster"

_cells, 2022, doi:10.3390/cells11091544_

Round 1
Reviewer 1 Report
Rafaella et al. has looked if the early serotonergic disturbance is correlative with PD onset and experimented to show that early Flx treatment ameliorates locomotor and anatomical functions.
Minor Comments:
- Use one form between pre-symptomatic and presymptomatic, three-days and three days, etc.
- Change “an early brief manipulation at line 26 “ to “a brief early.”
- Line 46, do you mean asymptomatic?
- Full form of abbreviations, e.g., MS at line 87 or FSCV at 315.
- Correct Sin SERT at line 290; perhaps you mean to say synaptic SERT
- Please change transcript to transcripts where required in the entire manuscript.
- Please correct “The same experiment was performed in younger flies” change “in” to “on”
- Usage of “The” and other articles is not correct. Please check your manuscript for English editing.
Major:
- Authors measured Tau in FSCV experiments and deduced that SERT activity is declined in Pink1B9. Then they fed these and controlled flies with Fluoxetine. They didn’t check whether three days of treatment during the presymptomatic phase was enough to restore SERT activity? This is an essential experiment, especially as the behavior data in Figure 3 is puzzling because control flies show the opposite effect. Also, could smaller fluoxetine dosages have reversed these results?
- In Figures 3B and 3C, the authors deduce the improvement in locomotor functions in fluoxetine treated Pink1B9 mutant by comparing Fluoxetine treated controls. However the behavior of Fluoxetine treated controls is surprising and not "remarkable" specifically as this is the only measurement where both data spread and hence error bar is very low; I would request authors to repeat this experiment, to show reproducibility.
- The authors used a one-time point of early three days and showed some locomotor improvements? Does this treatment enough to rescue non-motor impairment as well?
- It would have been great to show fluoxetine treatment at least two-time points: one early and one later symptomatic phase. This experiment is essential to claim that only early disturbance is causal.
- The more straightforward experiment would have been to perform the Pink1B mutants behavior experiments in SerT mutant background. Wonder why the authors do not choose the genetic approach. Alternatively, they could have also combined Pink1B mutants with Trh mutants.
- Lastly, the authors should discuss how Serotonin reaches PPL1 and PPL2 cells, are there any evidence of any 5-HT receptors localized to these dopaminergic cell populations?
Reviewer 2 Report
Zarete et al demonstrated that an early dysfunction in the serotonergic system precedes and contributes to the onset of the parkinsonian phenotype in Drosophila. The manuscript is very well written and is of immense significance to the field.
Author Response
We appreciate the reviewer’s comments. We are grateful that the reviewer has such a positive consideration on our work and considers that it advances the field.
We have checked the manuscript and made editions (highlighted in yellow in the new manuscript) when needed.
Reviewer 3 Report
The authors present here a study in a fly drosophila model of PD linked to PINK1 deletion showing interesting elements regarding serotonin neurotransmission during the presymptomatic phase of PD.
Overall I think this manuscript is worthy of publication. I have several changes suggestions and comments though, prior to publication acceptance:
- Fig 1: please add “in PINK1 mutant” regarding the presymptomatic and symptomatic phase as this shouldn’t apply to WT flies.
- A few remarks regarding stats and figures: The 2 way ANOVA with Bonferroni have been applied but it is not obvious that normality was reached regarding the results and that another test may have been necessary instead. Showing the SD bar, the bar graph are not very convincing as of to significance for some results (for example Figure 3 and 4 but also fig 2). Box plots could be presented instead. It could be interesting to see the results of normality test and variance comparison.
- minor English editing such as for example: Title of fig 2a: “SerT transcripts”, also page 10 line 391 replace by “it induced an increased number” or “it resulted in increased numbers” for instance
- Discussion could be improved by adding a few elements:
- could the authors provide elements regarding the known specificities of each subpopulations and difference of PPL1 and PPL2 DA neurons and why they could both be affected differently in these results.
- starting from line 465, could the authors further elaborate on how the modification of 5HT transmission (by acting on 5HT receptors) could modulate dopamine transmission (cf mechanism of action of specific atypical neuroleptics for instance).
- also comment on how the results of the present study add up to the fact that there are some arguments published on genetic forms of PD (such as Pink1 or Parkin) being more a development disorder rather than a neurodegenerative one.
Round 2
Reviewer 1 Report
The authors have revised, answered, and provided new information in response to my comments; now, the manuscript looks good and is ready to be published. I have no further comments. Best wishes: Farhan